# On the Cross-Graph Transferability of Dynamic Link Prediction

Submission Id: 2210

## ABSTRACT

Dynamic link prediction aims to predict the future links on dynamic graphs, which can be applied to wide scenarios such as recommender systems and social networks on the World Wide Web. Existing methods mainly (1) focus on the in-graph learning, which cannot generalize to graphs unobserved during training; or (2) achieve the cross-graph predictions in a many-many mechanism by training on multiple graphs across various domains, which results in a large computational cost. In this paper, we propose a cross-graph dynamic link predictor named CrossDyG, which achieves the cross-graph transferability in a one-many mechanism which trains on one single source graph and test on different target graphs. Specifically, we provide causal and empirical analysis on the structural bias caused by the graph-specific structural characteristics in cross-graph predictions. Then, we conduct deconfounded training to learn the universal network evolution pattern from one single source graph during training. Finally, we apply the causal intervention to leverage the graph-specific structural characteristics of each target graph during inference. Extensive experiments conducted on three benchmark data of dynamic graphs demonstrate that CrossDyG outperforms the state-of-the-art baselines by up to 11.01% and 17.02% in terms of AP and AUC, respectively. In addition, the improvements are especially significant when training on small source graphs. The implementation of our approach is available in https://anonymous.4open.science/r/CrossDyG-8B70.

## KEYWORDS

Dynamic Link Prediction; Network Science; Graph Learning

## 1 INTRODUCTION

Graph networks are effective tools for representing realistic complex systems on the World Wide Web like social networks [7, 17, 36], social media [25, 28] and recommender systems [23, 33, 37], where the contained elements are regarded as nodes and the interactions between them are deemed as edges, respectively [13, 39]. Moreover, graph networks usually continuously evolve in the real-world scenarios on the Web [15, 18, 27]. For instance, users build social connections with different friends on social networks at different timestamps, and users interact with items sequentially in recommender systems. Such temporal dynamics lead the graph networks to dynamic graphs, where the contained nodes and edges keep changing over time [3, 12]. As a fundamental task of forecasting the temporal network evolution, dynamic link prediction is widely investigated [2, 41, 43], which aims to predict the future links to appear in dynamic graphs.

Existing methods for dynamic link prediction mainly focus on the *in-graph setting* as explained in Fig. 1(a), where the evolution pattern is learned from the previously observed network (marked by solid lines) and then applied to the same network to predict its future links (marked by dashed lines). For instance, the literature [2, 24, 41] propose to model the temporal and structural

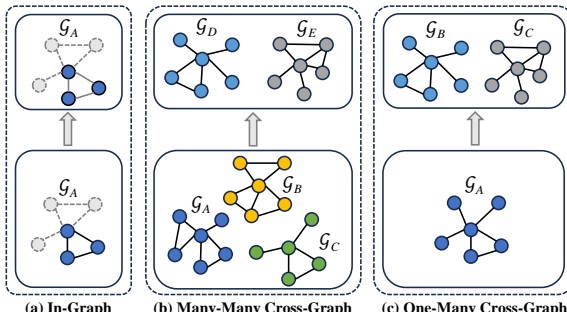

**(a) In-Graph**    **(b) Many-Many Cross-Graph**    **(c) One-Many Cross-Graph**

**Figure 1: Explanations of tasks associated with cross-graph transferability of dynamic link prediction, where different subscripts represent different graphs.**

neighborhood by including the node/edge features and the temporal features for dynamic representation learning. Moreover, the explicit structural correlations on the temporal topology are also taken into consideration using the neighbor co-occurrence mechanism of single hop [43, 45] or multiple hops [10, 35]. However, these models are generally designed for the in-graph scenarios, which **cannot generalize to other graphs unobserved in training**.

Recently, the dynamic graph foundation model has also been investigated to achieve cross-graph dynamic link prediction. For example, Huang et al. [8] propose DyExpert, which trains a decode-only Transformer on extensive dynamic graphs of various domains for achieving cross-graph transferability. That is, DyExpert follows a *many-many mechanism* as explained in Fig. 1(b), which learns the evolution pattern from the combination of multiple dynamic graphs, i.e., $\{\mathcal{G}_A, \mathcal{G}_B, \mathcal{G}_C\}$ and then applies the learned model on other graphs, i.e., $\mathcal{G}_D$ and $\mathcal{G}_E$. However, such a foundation model mechanism requires a large computational cost during both the data collection and model training stages, leading to a low efficiency in applications. Thus, it remains a challenging problem of how to **efficiently achieve the cross-graph dynamic link prediction**.

To solve the above-mentioned issues, we propose to investigate the cross-graph dynamic link prediction using a *one-many mechanism* as explained in Fig. 1(c), which trains on a single source graph $\mathcal{G}_A$ and test on different target graphs $\mathcal{G}_B$ and $\mathcal{G}_C$. However, considering that the structural characteristics in different dynamic graphs are diverse, there exist two main challenges: (1) **Challenge I:** *How to accurately learn the **universal network evolution pattern** across different graphs by eliminating the bias of structural characteristics on the single source dynamic graph during training?* (2) **Challenge II:** *How to effectively leverage the **graph-specific structural characteristics** in each target dynamic graph for making predictions during inference?*

To tackle the challenges, we propose a novel framework named Cross-graph Dynamic Graph link predictor (**CrossDyG**) for improving the cross-graph transferability of dynamic link prediction. Specifically, we first analyze how the structural characteristics introduce the bias in cross-graph dynamic link prediction based on both

causal analysis and empirical analysis. Then during the training stage, we perform deconfounded training to eliminate the structural bias on the single source dynamic graph, thus accurately capturing the universal network evolution pattern across different dynamic graphs. Finally, in the inference stage, based on the learned evolution pattern, we further adopt causal intervention to leverage the graph-specific structural characteristics of each target dynamic graph for making accurate predictions.

The main contributions in this paper are summarized as follows:

(1) To the best of our knowledge, we are the first to investigate the cross-graph dynamic link prediction in a one-many mechanism, and study the impact of structural bias on the cross-graph transferability based on causal analysis and empirical analysis.

(2) We adopt the deconfouded training to learn the universal network evolution pattern across different graphs from one single source graph in training, and design causal intervention strategy to leverage the graph-specific structural characteristics of each target graph for accurate predictions during inference.

(3) Extensive experiments conducted on three real-world dynamic graphs present the superiority of CrossDyG against the state-of-the-art baselines, where the performance gains are up to 11.01% and 17.02% in terms of AP and AUC, respectively.

## 2 RELATED WORK

### 2.1 Dynamic Link Prediction

As an important foundation of predicting temporal network evolution, dynamic link prediction is widely investigated recently. Specifically, most methods focus on the in-graph learning which learns the network evolution from the historically observed graph and then predicts future links in the same graph.

Early methods mainly focus on generating dynamic node representations by learning from the temporal features and node/edge features [2, 24, 41, 42]. For instance, Xu et al. [41] introduce the temporal graph attention layer, which efficiently considers the temporal and topological neighborhood information using the self-attention mechanism [32]. And Cong et al. [2] design a conceptually and technically simple architecture GraphMixer which summarizes features from one-hop temporal neighbors using the MLP-mixer [30].

Moreover, some recent literatures [10, 29, 35, 43] propose to leverage the structural correlations between nodes on the temporal structural topology. Specifically, Wang et al. [35] first design a causal anonymous walk strategy to automatically extract the temporal network motifs for considering the node correlations. And Yu et al. [43] simplify the causal anonymous walk by introducing a simple but effective neighbor co-occurrence mechanism to learn from the historical neighbor sequence, i.e., one-hop neighbors.

In addition to the above in-graph learning methods, a recent dynamic graph foundation model DyExpert is designed to achieve the cross-graph transferability [8]. Specifically, DyExpert follows a many-many mechanism which learns the universal network evolution pattern from multiple graphs of various domains, and then applies the learned link predictor on other graphs.

However, on the one hand, the in-graph dynamic link prediction methods cannot generalize to other graphs which are not observed during the training stage. And on the other hand, though the current dynamic foundation model achieves the cross-graph

predictions, its many-many transferability strategy requires large computational cost in both the data collection and model training stages. Thus, we investigate the cross-graph transferability in a one-many mechanism, which still remains a challenging problem.

### 2.2 Causal Inference

Causal inference [20] constitutes a field of study dedicated to quantifying the causal connections among variables, which is widely adopted to eliminate the negative bias in various real-world applications, such as recommender systems [26, 38, 44] and graph analysis [4, 16, 19]. For example, Wei et al. [38] design a model-agnostic counterfactual reasoning framework which uses multi-task learning to obtain the contribution of difference causes to the user-item interactions, and then perform counterfactual inference to remove the item popularity bias during inference. Moreover, Zhang et al. [44] propose to conduct deconfounded learning using $do$-calculus to remove the popularity bias during training and then causally intervene the predicted future item popularity for inference. In addition, Fan et al. [4] introduce a general disentangling graph neural network (GNN) framework to learn the separate causal substructure and bias substructure for generating the counterfactual unbiased samples, thus improving the generalization ability of GNNs.

Inspired by the above works, we utilize deconfounded learning to eliminate the structural bias to learn the universal network evolution pattern from one single source graph, and further adopt causal intervention to leverage the graph-specific structural characteristics for predictions on other dynamic graphs during inference.

## 3 APPROACH

### 3.1 Problem Statement

*Dynamic graph.* We first give the definition of dynamic graph, which can be denoted as $\mathcal{G} = \{\mathcal{V}, \mathcal{E}\}$. Here $\mathcal{V}$ is the node set, the size of which continuously increases with time in dynamic graphs, and $\mathcal{E}$ is the edge set, each edge $(u, v, t) \in \mathcal{E}$ connects nodes $u$ and $v$ at the timestamp $t$. In addition, each node and each edge is associated with a node feature $\mathbf{x}_u \in \mathbb{R}^{d_N}$ and an edge feature $\mathbf{e}_{uv}^t \in \mathbb{R}^{d_E}$, respectively, where $d_N$ and $d_E$ are the respective dimensions of node feature and edge feature.

*In-graph dynamic link prediction.* Existing dynamic graph models generally concentrate on the in-graph dynamic link prediction. Specifically, given a timestamp $t'$, the observed dynamic graph can be denoted as $\mathcal{G}' = \{\mathcal{V}', \mathcal{E}'\}$, where $\mathcal{V}'$ is the node set containing nodes appearing before $t'$ and $\mathcal{E}'$ is the edge set consisting of edges happening before $t'$. In-graph dynamic link prediction aims to learn the temporal network evolution pattern from $\mathcal{G}' = \{\mathcal{V}', \mathcal{E}'\}$ and then predicts each link with timestamp $t > t'$, i.e., $(u, v, t) \in \mathcal{E} \setminus \mathcal{E}'$.

*Cross-graph dynamic link prediction.* Different from the in-graph setting, our research focuses on a more challenging task: cross-graph dynamic link prediction. Specifically, we aim to train a dynamic link predictor $f_\theta$ to learn the universal network evolution pattern from source dynamic graph $\mathcal{G}_s = \{\mathcal{V}_s, \mathcal{E}_s\}$, and then apply the learned model $f_\theta$ to target dynamic graph $\mathcal{G}_t = \{\mathcal{V}_t, \mathcal{E}_t\}$.

### 3.2 System Overview

Given source node $u$ and target node $v$ in dynamic graph $\mathcal{G} = \{\mathcal{V}, \mathcal{E}\}$, we predict the probability of forming a future link between $u$ and $v$ at the timestamp $t$ by the following three steps:

### 3.2.1 Neighbor sequence sampling.

Given the future link $(u, v, t)$ to predict, the neighbor sequence is first sampled for both source node $u$ and target node $v$ by backtracking the interaction timestamps. For example, the sampled neighbor sequence for source node $u$ can be denoted as $\mathcal{N}_u = \{(v_1, t_1), \ldots, (v_K, t_K)\}$, where $K$ is the sampled neighbor number and $t_1 < \cdots < t_K < t$.

### 3.2.2 Dynamic representation learning.

After that, the dynamic representations of nodes $u$ and $v$ are generated by combining the structural encoding, the temporal encoding and the node/edge features. Here we take node $u$ as an example for illustrations.

*Structural encoding.* Neighbor co-occurrence mechanism (NCM), which we denote as $f_\psi$, is a commonly adopted strategy for encoding the node structures in dynamic link prediction. Specifically, given node $u$ with neighbors $\mathcal{N}_u$ and node $v$ with neighbors $\mathcal{N}_v$, $f_\psi$ counts the occurrence frequency of each node $v_i \in \mathcal{N}_u \cup \mathcal{N}_v$ in $\mathcal{N}_u$ and $\mathcal{N}_v$, respectively. This can be formulated as follows:

$$f_\psi(v_i) = [g(v_i, \mathcal{N}_u), g(v_i, \mathcal{N}_v)], \quad (1)$$

where $f_\psi(v_i) \in \mathbb{R}^2$ is the structural encoding vector of neighbor $v_i$, and $g$ indicates the function counting the occurrence frequency of $v_i$ in $\mathcal{N}_u$ as $g(v_i, \mathcal{N}_u) = |\{v_j | v_j \in \mathcal{N}_u, v_j = v_i\}|$. For instance, given $\mathcal{N}_u = \{v_1, v_2, v_2, v_4\}$ and $\mathcal{N}_v = \{v_1, v_1, v_2, v_3\}$, $f_\psi(v_1) = [1, 2]$, $f_\psi(v_2) = [2, 1]$, $f_\psi(v_3) = [0, 1]$ and $f_\psi(v_4) = [1, 0]$. Correspondingly, the structural encoding of $u$ and $v$ are $[[1, 2], [2, 1], [2, 1], [1, 0]]^\mathsf{T}$ and $[[1, 2], [1, 2], [2, 1], [0, 1]]^\mathsf{T}$, respectively.

Then, the neighbor co-occurrence mechanism $f_\psi$ and a multi-layer perceptrons (MLP) which we denote as $f_\phi$ are combined to encode the structural features of each neighbor $v_i \in \mathcal{N}_u$ as follows:

$$\mathbf{s}_i = f_\phi(f_\psi(v_i)) = \mathrm{MLP}(f_\psi(v_i)), \quad (2)$$

where $\mathbf{s}_i \in \mathbb{R}^d$ is the generated structural embedding of $v_i$.

*Temporal encoding.* Moreover, a continuous time encoding function [40] is adopted for temporal encoding, to model the time interval $\Delta t_i = t - t_i$ between the timestamp $t$ and the interaction timestamp $t_i$ with neighbor $v_i$, which can be formulated as follows:

$$\mathbf{t}_i = [cos(\omega_1 \Delta t_i), sin(\omega_1 \Delta t_i), \ldots, cos(\omega_d \Delta t_i), sin(\omega_d \Delta t_i)], \quad (3)$$

where $[\omega_1, \ldots, \omega_d]$ are the learnable parameters.

*Information fusion.* After that, we generate the fused embeddings of $u$'s neighbors, for example, the representation of neighbor $v_i \in \mathcal{N}_u$ can be generated by combining the structural embedding $\mathbf{s}_i$, the temporal embedding $\mathbf{t}_i$, the node features $\mathbf{x}_i$ and the edge features $\mathbf{e}_i$, which can be formulated as $\mathbf{z}_i = [\mathbf{s}_i^u, \mathbf{t}_i^u, \mathbf{x}_i^u, \mathbf{e}_i^u] \in \mathbb{R}^{2d + d_N + d_E}$.

### 3.2.3 Prediction and model optimization.

Finally, predictions are made by fusing the representations of neighbors of $u$ and $v$, i.e., $\mathbf{Z}_u, \mathbf{Z}_v \in \mathbb{R}^{K \times (2d + d_N + d_E)}$, by inputting them to Transformer [32], MLP-Mixer [30], etc, for obtaining the prediction score as $y_{uv}^t$. Then, we adopt cross-entropy as the loss function for model optimization:

$$L = \sum_{(u,v,t) \in \mathcal{E}_s} -\log(\sigma(y_{uv}^t)) - \log(\sigma(1 - y_{un}^t)), \quad (4)$$

where $(u, v, t) \in \mathcal{E}_s$ denotes each temporal edge in the source dynamic graph $\mathcal{G}_s = \{\mathcal{V}_s, \mathcal{E}_s\}$, and $y_{un}^t$ is the prediction score of the randomly sampled negative edge $(u, n, t)$.

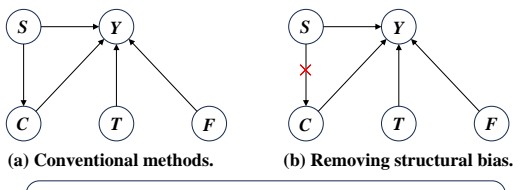

(a) Conventional methods.  (b) Removing structural bias.

S: Structural characteristics;  C: Structural Correlations;
T: Temporal characteristics;  F: Node/edge features;
Y: Prediction probability.

**Figure 2: Causal graphs for (a) the conventional methods and (b) the version removing the structural bias.**

## 3.3 Analysis of Structural Bias

### 3.3.1 Effect of Structural Bias from a Causal View.

To clearly understand the bias caused by the structural characteristics (i.e., $f_\psi(v_i)$ obtained by Eq. (1)) on the prediction probability, we adopt the causal graph [20], which denotes variables as nodes and describe relations between them as edges. Here we construct the causal graph of conventional dynamic graph models and our improved version which removes the structural bias, which are presented in Fig. 2. As shown in Fig. 2, there are five variables, including: (1) $S$ denotes the **s**tructural characteristics of nodes; (2) $C$ denotes the structural **c**orrelations between nodes; (3) $T$ denotes the **t**emporal characteristics of nodes; (4) $F$ denotes the node/edge **f**eatures; (5) $Y$ denotes the probability of future link formation.

Then, we can observe that there exist two main causal relations in conventional dynamic models as shown in Fig 2(a) as follows:

- $\{S, C, T, F\} \rightarrow Y$ denotes that the probability of forming a future link $(u, v, t)$ is determined by the combination of the structural characteristics $S$, the structural correlations $C$, the temporal characteristics $T$ and the node/edge features $F$;
- $S \rightarrow C \rightarrow Y$ denotes that learning the structural correlations $C$ is influenced by the structural characteristics $S$ and then affects the prediction probability $Y$;

Moreover, from the causal graph, we can observe that there exist two causal paths from the structural characteristics $S$ to the prediction probability $Y$, i.e., $S \rightarrow C \rightarrow Y$ and $S \rightarrow Y$. The first path influences the accurate estimation of the structural correlations $C$'s impact on the prediction probability $Y$, thus cannot accurately learn the ***universal network evolution pattern (i.e., $\{C, T, F\} \rightarrow Y$)*** across different dynamic graphs. The second path indicates that ***the graph-specific structural characteristics*** in each graph also affects the prediction probability (i.e., $S \rightarrow Y$), which is expected when applied in one specific dynamic graph.

### 3.3.2 Analysis on conventional methods.

Existing methods suffers from the structural bias $S$ when learning the universal network evolution pattern $\{C, T, F\} \rightarrow Y$. Specifically, the conditional probability $P(Y|C, T, F)$ in conventional methods can be calculated as:

$$\begin{aligned}
P(Y|C, T, F) &\stackrel{(1)}{=} \sum_s P(Y, s|C, T, F), \\
&\stackrel{(2)}{=} \sum_s P(Y|C, T, F, s)P(s|C, T, F), \\
&\stackrel{(3)}{=} \sum_s P(Y|C, T, F, s)P(s|C), \\
&\stackrel{(4)}{\propto} \sum_s P(Y|C, T, F, s)P(C|s)P(s).
\end{aligned} \quad (5)$$



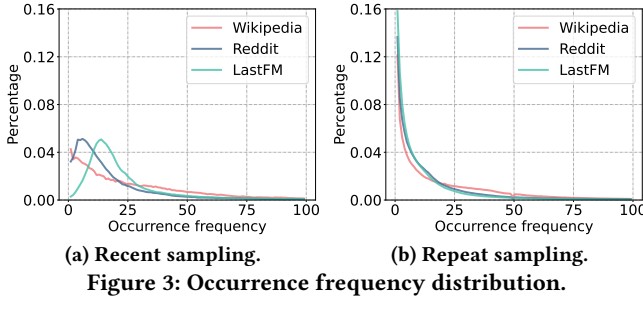

(a) Recent sampling.  (b) Repeat sampling.

Figure 3: Occurrence frequency distribution.

where (1) is the definition of marginal distribution; (2) is because of the Bayes' theorem; (3) is because $T$ and $F$ are independent to $S$ according to the causal graph; (4) is because of Bayes' theorem. Then, we can observe that there exists a term $P(C|S)$, which indicates the influence of the structural characteristics $S$ on the structural correlations $C$, and thus affecting the prediction probability $Y$.

*3.3.3 Empirical analysis.* In addition, we further explain why the structural bias exists in cross-graph dynamic link prediction using empirical analysis. Specifically, we plot the occurrence frequency (i.e., $f\psi(v_i)$ generated by Eq. (1)) of neighbors obtained by the commonly adopted recent sampling [43] (i.e., selecting the most recent interacted neighbors) in Fig. 3a. It can be observed that the occurrence frequency distributions obtained by recent sampling are obviously different across three graphs. Such differences lead to the unsatisfactory performance of existing dynamic graph models on cross-graph dynamic link prediction, since the learned model trained to fit the distribution of source dynamic graph fails to generalize to target dynamic graph. Through the above analysis, we can conclude that the bias of structural characteristics causes negative effect on the cross-graph transferability of dynamic link prediction.

*3.3.4 Repeat neighbor sampling.* To better model the uniformity of different dynamic graphs from improving the cross-graph trasferability, we adopt repeat sampling [45] for neighbor extraction in our proposal. Specifically, given source node $u$ and target node $v$ and their neighbors as $\mathcal{N}_u$ and $\mathcal{N}_v$, for node $u$, we sample the historical neighbors equal to target node $v$, i.e., $\{v_i \in \mathcal{N}_u | v_i = v\}$. For instance, given source node $v_1$, target node $v_2$ and all historical neighbors of $v_1$ as $\{(v_2, t_1), (v_3, t_2), (v_2, t_3), (v_4, t_4)\}$, we extract the neighbor sequence of $v_1$ as $\{(v_2, t_1), (v_2, t_3)\}$. In addition, if target node $v$ does not appear in the historical neighbors of source node $u$, then the neighbors of $u$ are sampled using the recent sampling. And the neighbors of target node $v$ are obtained in the same way.

Through the recent neighbor sampling, we can extract the neighbors most related to the node structural correlations by removing the noisy neighbors. To clearly illustrate this, we plot the occurrence frequency distributions of neighbors obtained by the repeat sampling in Fig. 3b, from which we can observe that the distributions of three dynamic graphs obtained by repeat sampling are obviously closer than recent sampling. Such close distributions largely contribute to learning the cross-graph transferability.

However, it can also be observed that there still exist differences between the occurrence frequency distributions of three dynamic graphs obtained by repeat sampling, which is caused by the graph-specific structural characteristics in different graphs. Thus, we propose CrossDyG, which resorts to deconfounded training to learn

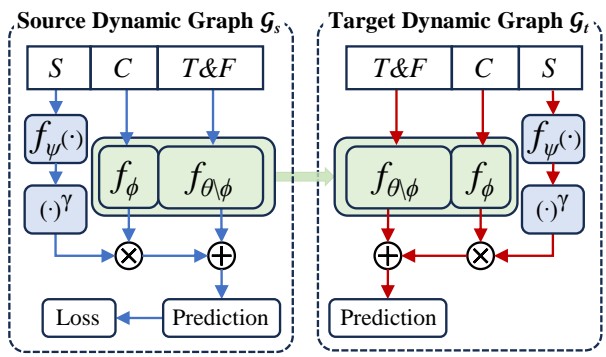

Figure 4: Pipeline of CrossDyG, where the blue and red lines denote the training and inference stages, respectively.

the ***universal network evolution pattern*** across different graphs by removing structural bias during training (see Section 3.4) on source graph and then utilize the ***graph-specific structural characteristics*** for making predictions during inference on target graph (see Section 3.5). The pipeline of CrossDyG is plotted in Fig. 4.

## 3.4 Deconfounded Training

To achieve our aim that removes the structural bias $S$ in learning $\{C, T, F\} \rightarrow Y$, we adopt the *do*-calculus [20] in causal science, which forces to remove the impact of $\{C, T, F\}$'s parent nodes, i.e., $C$'s patent node $S$. Then, let the causal graphs in Fig. 2(a) and 2(b) be $G$ and $G'$, respectively, $P(Y|do(C, T, F))$ is calculated as:

$$
\begin{aligned}
P(Y|do(C,T,F)) &\overset{(1)}{=} P_{G'}(Y|C,T,F), \\
&\overset{(2)}{=} \sum_s P_{G'}(Y|C,T,F,s)P_{G'}(s|C,T,F), \\
&\overset{(3)}{=} \sum_s P_{G'}(Y|C,T,F,s)P_{G'}(s), \\
&\overset{(4)}{=} \sum_s P(Y|C,T,F,s)P(s).
\end{aligned}
\tag{6}
$$

where $P_{G'}(\cdot)$ indicates the probability evaluated on $G'$. (1) is because of backdoor criterion [20] as the backdoor paths of $C \leftarrow S \rightarrow Y$ in $G$ have been blocked by $do(C, T, F)$; (2) is because of Bayes' theorem; (3) is because of that $\{C, T, F\}$ are independent with $S$ in $G'$; (4) is because that the causal mechanism $\{S, C, T, F\} \rightarrow Y$ is not changed when cutting off $S \rightarrow C$, $P(S) = P_{G'}(S)$ since $S$ have the same prior on the two graphs.

Comparing Eq. (6) with Eq. (5), we can find that the term $P(C|S)$ in Eq. (5) disappears. This indicates that by applying the *do*-calculus on the variables $\{C, T, F\}$, i.e., $C$, we can obtain the debiased prediction probability as $P(Y|do(C, T, F)) = \sum_s P(Y|C, T, F, s)P(s)$. Specifically, this eliminates the impact of the bias of structural characteristics (i.e., $\{S \rightarrow Y\}$) in source dynamic graph during training, thus learning the ***universal network evolution pattern*** (i.e., $\{C, T, F\} \rightarrow Y$) across different dynamic graphs. Next, we aim to estimate $P(Y|do(C, T, F))$ from the training data on source dynamic graph $\mathcal{G}_s = \{\mathcal{V}_s, \mathcal{E}_s\}$, including two steps, i.e., estimating $P(Y|S, C, T, F)$ and $P(Y|C, T, F, s)P(s)$, respectively.

*Step 1: Estimating $P(Y|S, C, T, F)$.* This conditional probability function evaluates that given a node pair $(u, v, t)$ with node correlations as $C = c$, temporal characteristics as $T = t$, node/edge features as $F = f$ and structural characteristics as $S = s$, how is the probability $P(y = 1|s, c, t, f)$ of forming a future link between $u$ and $v$ at the timestamp $t$. This can be implemented by the dynamic graph models with the learnable parameters $\theta$ introduced in Section 3.2, where the parameters $\theta$ can be optimized by the cross-entropy loss formulated by Eq. (4).

Moreover, given the structural characteristics as $s$, to learn the universal network evolution pattern, we propose to decouple the structural characteristics $s$ from the modeling processing of predicting $Y$, i.e., $P_\theta(y = 1|c, t, f, s)$. In addition, considering that $T$ and $F$ are independent from $S$ in $G'$ as shown in Fig. 3(b), the variables we need to decouple are the structural correlations $C$ and the structural characteristics $S$, which are together modeled by the structural encoding function in Eq. (2). Specifically, to achieve such an aim of decoupling $C$ and $S$, we design a decoupled structural encoding function by improving Eq. (2) as follows:

$$f_\phi(f_\psi(v_i)) = f_\phi(H(f_\psi(v_i))) \times (I(f_\psi(v_i)))^\gamma, \quad (7)$$

where $\gamma$ is the parameter for smoothing the occurrence frequency, $I(\cdot)$ is the Identity Function, and $H(\cdot)$ denotes the Unit Step Function defined as follows:

$$H(x) = \begin{cases} 1 & \text{if } x > 0, \\ 0 & \text{if } x \leq 0. \end{cases} \quad (8)$$

Here the former part $f_\phi(H(f_\psi(v_i)))$ indicates that only neighbor $v_i$'s appearances in $\mathcal{N}_u$ and $\mathcal{N}_v$ (i.e., if the occurrence happens or not) are inputted to the MLP $f_\phi$, while the occurrence frequency is ignored. That is, $f_\phi(H(f_\psi(v_i)))$ is decoupled with the latter part $(I(f_\psi(v_i)))^\gamma$ which records the occurrence frequency of $v_i$ in $\mathcal{N}_u$ and $\mathcal{N}_v$. For instance, given $\mathcal{N}_u = \{v_1, v_2, v_2, v_2\}$ and $\mathcal{N}_v = \{v_1, v_2, v_3, v_3\}$, $H(f_\psi(v_2)) = [1, 1]$, $H(f_\psi(v_3)) = [0, 1]$, while $I(f_\psi(v_2)) = [3, 1]$, $I(f_\psi(v_3)) = [0, 2]$.

*Step 2: Estimating $P(Y|C, T, F, s)P(s)$.* Then we proceed to estimate the interventional probability $P(Y|do(C, T, F))$. Considering that the space of $S$ is extremely large, it is impractical to perform a summation over its space for making predictions. Fortunately, we can employ the following reduction to eliminate the need for this summation as follows:

$$
\begin{aligned}
P(Y|do(C, T, F)) &\stackrel{(1)}{=} \sum_s P(Y|C, T, F, s)P(s), \\
&\stackrel{(2)}{=} f_{\theta\backslash\phi}(t, f) + \sum_s f_\phi(c) \times s^\gamma P(s), \\
&\stackrel{(3)}{=} f_{\theta\backslash\phi}(t, f) + f_\phi(c) \sum_s s^\gamma P(s), \\
&\stackrel{(4)}{=} f_{\theta\backslash\phi}(t, f) + f_\phi(c)E(S^\gamma).
\end{aligned}
\quad (9)
$$

where $E(S^\gamma)$ denotes the expectation of $S^\gamma$, which is a constant. Moreover, considering that the value $E(S^\gamma)$ is equal for all node pairs in the same dynamic graph, the existence of $E(S^\gamma)$ does not change the results when comparing the prediction probability of different links in a single graph. Thus, we can adopt the learned model with parameters $\theta$ including $f_{\theta\backslash\phi}$ and $f_\phi$ to estimate $P(Y|do(C, T, F))$.

To sum up, in the deconfounded learning, we train the learned model with parameters $\theta$ by fitting the links in source dynamic graph, and then use $f_{\theta\backslash\phi}$ and $f_\phi$ contained in the learned model to capture the universal network evolution pattern, i.e., $\{C, T, F\} \to Y$ by removing the negative impact of structural bias, i.e., $C \to Y$.

### 3.5 Structural Intervention in Inference

Through calculating the probability $P(Y|do(C, T, F))$, we can eliminate the negative effect of the structural bias, thus accurately learn the **universal network evolution pattern** across different graphs. Then, after training on source dynamic graph $\mathcal{G}_s = \{\mathcal{V}_s, \mathcal{E}_s\}$, in the inference stage of predicting links in target dynamic graph $\mathcal{G}_t = \{\mathcal{V}_t, \mathcal{E}_t\}$, we propose to leverage the **graph-specific structural characteristics** for better fitting in the target graph. Specifically, given the structural characteristics of target graph as $S = \tilde{s}$, this can be achieved by the intervention $do(S = \tilde{s})$ as:

$$
\begin{aligned}
&P(Y|do(C = c, T = t, F = f), do(S = \tilde{s})) \\
&= P_\theta(y = 1|c, t, f, \tilde{s}), \\
&= f_{\theta\backslash\phi}(t, f) + f_\phi(H(f'_\psi(v_i))) \times (I(f'_\psi(v_i)))^\gamma.
\end{aligned}
\quad (10)
$$

where $f'_\psi(v_i)$ indicates the graph-specific structural characteristics, i.e., the occurrence frequency, in target graph $\mathcal{G}_t$. In addition, in our improved causal graph $G'$ shown in Fig. 2(b), the above calculated result directly equals to the conditional probability $P(Y|S, C, T, F)$, since there is no backdoor path between $S$ and $C$ in $G'$.

## 4 EXPERIMENTS

### 4.1 Research questions

We validate the effectiveness of our proposed CrossDyG by answering the following five research questions:

**RQ1** Can CrossDyG improve the cross-graph transferability across different graphs in a one-many mechanism by learning from one single source dynamic graph?

**RQ2** How is the contribution of different contained components in CrossDyG to the prediction performance?

**RQ3** How is the performance of CrossDyG compared with the baselines when training on multiple source dynamic graphs?

**RQ4** Can CrossDyG mitigate the bias of structural characteristics in training and leverage them in inference, respectively?

**RQ5** How is the sensitivity of CrossDyG's performance to the contained hyper-parameters?

### 4.2 Experimental Settings

*4.2.1 Datasets and evaluation metrics.* To validate the effectiveness of CrossDyG, we evaluate the prediction performance of CrossDyG and the baselines on three real-world dynamic graphs of different sizes, *i.e.*, Wikipedia, Reddit and LastFM, which are widely adopted in dynamic link prediction [2, 14, 35].

- **Wikipedia** is a temporal interaction graph where the contained nodes represent the Wikipedia editors and the pages they modify, and edges signify the timestamped editing actions over one month. In addition, each edge is associated with a 172-dimensional Linguistic Inquiry and Word Count (LIWC) feature.
- **Reddit** is a temporal interaction graph recording the activity of users engaging with different subreddits over a monthly period. Nodes in Reddit include users and subreddits, and edges represent

**Table 1: Performance of CrossDyG and the baselines, where we adopt each of Wikipedia, Reddit and LastFM as Source for training and utilize each other dataset as Target for evaluation. The best performer and the best baseline in each line are bold and underlined, respectively, and "Improve." indicates the improvement percentage of CrossDyG above the best baseline.**

| Metric | Source | Wikipedia | | Reddit | | LastFM | |
|---|---|---|---|---|---|---|---|
| | Target | Reddit | LastFM | Wikipedia | LastFM | Wikipedia | Reddit |
| AP | TGAT | 0.7894±0.0124 | 0.6103±0.0020 | 0.8641±0.0058 | 0.5356±0.0115 | 0.9022±0.0083 | 0.5764±0.0125 |
| | TCL | 0.7125±0.0154 | 0.6182±0.0052 | 0.8178±0.0135 | 0.5403±0.0045 | 0.9393±0.0135 | 0.5377±0.0118 |
| | GraphMixer | 0.9055±0.0032 | 0.6935±0.0034 | 0.9277±0.0029 | 0.6999±0.0052 | 0.9127±0.0066 | 0.7496±0.1121 |
| | CAWN | 0.8711±0.0056 | 0.7460±0.0101 | 0.9499±0.0037 | 0.8196±0.0099 | 0.9368±0.0053 | 0.9298±0.0087 |
| | DyGFormer | 0.8961±0.0083 | 0.7537±0.0318 | 0.9578±0.0045 | **0.8305±0.0128** | 0.9511±0.0069 | 0.9354±0.0138 |
| | CrossDyG | **0.9532±0.0083** | **0.8367±0.0072** | **0.9616±0.0031** | 0.8453±0.0047 | **0.9650±0.0033** | **0.9525±0.0016** |
| | Improve. | 5.27% | 11.01% | 0.40% | -0.22% | 1.46% | 1.79% |
| AUC | TGAT | 0.8055±0.0057 | 0.6058±0.0015 | 0.8682±0.0033 | 0.5464±0.0091 | 0.8898±0.0098 | 0.5870±0.0146 |
| | TCL | 0.7137±0.0191 | 0.6042±0.0045 | 0.8230±0.0078 | 0.5542±0.0052 | 0.9240±0.0171 | 0.5248±0.0089 |
| | GraphMixer | 0.8997±0.0038 | 0.6928±0.0034 | 0.9247±0.0026 | 0.7088±0.0042 | 0.9043±0.0073 | 0.7627±0.1016 |
| | CAWN | 0.8560±0.0061 | 0.7117±0.0112 | 0.9325±0.0042 | 0.8250±0.0103 | 0.9083±0.0089 | 0.9176±0.0094 |
| | DyGFormer | 0.8877±0.0075 | 0.7311±0.0305 | 0.9435±0.0078 | 0.8319±0.0193 | 0.9323±0.0134 | 0.9210±0.0210 |
| | CrossDyG | **0.9503±0.0055** | **0.8555±0.0092** | **0.9602±0.0031** | **0.8666±0.0037** | **0.9648±0.0022** | **0.9481±0.0014** |
| | Improve. | 5.62% | 17.02% | 1.77% | 4.17% | 3.49% | 2.94% |

the posting requests with timestamps, where each edge contains a 172-dimensional LIWC feature vector.

- **LastFM** is a temporal graph that captures the user-song interactions over a one-month window, where users as well as songs are represented as nodes and the behaviors of users listening to songs are regarded as edges, respectively.

The statistics of three adopted datasets are shown in Table 2. In addition, following [35, 41, 43], average precision (AP) and area under the ROC curve (AUC) are adopted as the evaluation metrics.

*4.2.2 Baselines for comparison.* We select five state-of-the-art baselines to compare with CrossDyG for validating its superiority, including TGAT [41], TCL [34], GraphMixer [2], CAWN [35] and DyGFormer [43].

- **TGAT** [41] uses self-attention [32] and a continuous time encoding technique [40] based on the classical Bochner's theorem [22] to learn time-dependent node embeddings for predictions.
- **TCL** [34] designs a topology-aware Transformer containing a two-stream encoder with co-attention for modeling the semantic-level inter-dependencies, and introduces contrastive learning [6] to maximize mutual information between future interaction nodes.
- **GraphMixer** [2] designs a simple architecture for temporal graph learning that uses multi-layer perceptrons (MLP) and mean pooling for link and node encoding, respectively, followed by an MLP-based link classifier for making predictions.
- **CAWN** [35] designs a causal anonymous walk method which uses temporal random walks to extract motifs that represent network dynamics, with an anonymization strategy replacing the node identities with the hitting counts.
- **DyGFormer** [43] is a Transformer-based dynamic graph learning method that uses a neighbor co-occurrence mechanism and a patching technique to capture nodes' correlations and long-term temporal dependencies, respectively.

**Table 2: Statistics of the datasets used in our experiments.**

| Statistics | #Nodes | #Links | #N&L Feat | Unique Steps |
|---|---|---|---|---|
| Wikipedia | 9,227 | 157,474 | −&172 | 152,757 |
| Reddit | 10,984 | 672,447 | −&172 | 669,065 |
| LastFM | 1,980 | 1,293,103 | −&− | 1,283,614 |

It is worth noting that several memory networks based methods such as JODIE [14], DyRep [31] and TGN [24] are not included in comparison since their designs are not applicable to cross-graph link prediction. In addition, the dynamic graph foundation model DyExpert [8], which achieves the cross-graph transferability in a many-many mechanism, is not directly included because of the unfair comparison, since it adopts multiple dynamic graphs from various domains for training. Instead in RQ3, we compare the performance of CrossDyG and the baselines by training on multiple source dynamic graphs by following the setting in DyExpert.

*4.2.3 Model implementation.* We implement our proposed CrossDyG and the baselines based on the open-source toolkit proposed in [43]. Specifically, we adopt Adam with a learning rate of $1e^{-4}$ for model optimization, where the batch size is set to 200 and the dimensions of structural and temporal encodings are both set to 100. Moreover, different from existing works [21, 41, 43] which separate each dynamic graph to different parts for training and evaluation, we directly adopt each full dynamic graph for training or evaluation. It is worth noting that considering there is no validation data for early stopping in model training, we tune the number of training epochs a hyper-parameter. Specifically, we search the training epoch on source dynamic graph in $\{1, 2, \ldots, 10\}$. In addition, we tune the scaling number $\gamma$ in $\{0.1, 0.2, \ldots, 1.0\}$, the sampled neighbor number in $\{16, 32, \ldots, 160\}$, respectively, to find the optimal performance on each dataset. Furthermore, all experimental results are implemented using PyTorch 1.8.1 on a Ubuntu 18.04 server with NVIDIA GeForce RTX 3090 GPU with 24 GB memories.

**Table 3: Ablation study on different designed components in CrossDyG, where the performance decreasing percent of each variant is shown below its performance, and the biggest drop in each column is marked using the symbol ↓.**

| Metric | Source | Wikipedia | | Reddit | | LastFM | |
|---|---|---|---|---|---|---|---|
| | Target | Reddit | LastFM | Wikipedia | LastFM | Wikipedia | Reddit |
| AP | CrossDyG | 0.9532±0.0083 | 0.8367±0.0072 | 0.9616±0.0031 | 0.8453±0.0047 | 0.9650±0.0033 | 0.9525±0.0016 |
| | w/o Repeat | 0.6929±0.0106↓ | 0.5452±0.0096↓ | 0.7614±0.0256↓ | 0.5254±0.0220↓ | 0.7757±0.0024 | 0.5511±0.0241↓ |
| | w/o Causal | 0.8023±0.0415 | 0.6053±0.0348 | 0.9282±0.0073 | 0.8560±0.0013 | 0.5801±0.0094↓ | 0.6929±0.0107 |
| AUC | CrossDyG | 0.9503±0.0055 | 0.8555±0.0092 | 0.9602±0.0031 | 0.8666±0.0037 | 0.9648±0.0022 | 0.9481±0.0014 |
| | w/o Repeat | 0.7030±0.0117↓ | 0.5483±0.0100↓ | 0.7783±0.0138↓ | 0.5349±0.0197↓ | 0.7487±0.0026 | 0.5794±0.0317↓ |
| | w/o Causal | 0.8341±0.0507 | 0.6447±0.0387 | 0.9505±0.0020 | 0.8742±0.0014 | 0.3992±0.0066↓ | 0.6762±0.0251 |

## 4.3 Overall Performance (RQ1)

We compare the performance of CrossDyG and the baselines in terms of AP and AUC by training on each of Wikipedia, Reddit and LastFM and test on each other dataset. The results are presented in Table 1, from which we can obtain the following observations:

- ***Dynamic link prediction which reflects the network evolution is generalizable across different graphs.*** First, we can observe that the baselines which are designed for in-graph setting can also achieve satisfactory performance on the cross-graph dynamic link prediction. Moreover, comparing with the methods TGAT, TCL and GraphMixer which model the temporal characteristics and the node/edge features, the structural correlations aware models CAWN and DyGFormer obviously improve the performance. These observations indicate that various features on dynamic graphs which represent the network evolution pattern is universal across different graphs and thus can be generalizable.

- ***CrossDyG effectively improves the performance of the cross-graph dynamic link prediction task.*** Moreover, compared with the competitive baselines, we can observe that our proposed CrossDyG generally achieves the state-of-the-art performance in terms of AP and AUC on various scenarios transferring across different dynamic graphs. This benefits from our novel designs which utilize deconfounded learning to learn the universal network evolution pattern during training and adopt causal intervention to utilize the graph-specific structural characteristics in each target dynamic graph during inference.

- ***The advantages of CrossDyG are especially obvious when learning from small dynamic graphs.*** In addition, it is worth noting that the improvements of CrossDyG above the baselines are especially obvious when training on small dynamic graphs. For example, on the AP metric, CrossDyG outperforms the best baselines by 5.27% and 11.01% when training on the smallest graph Wikipedia, while the improvements are 0.40%, -0.22% and 1.46%, 1.79% when training on Reddit and LastFM, respectively. This specifies the strong ability of CrossDyG to learn the cross-graph transferability across dynamic graphs from limited data.

## 4.4 Ablation Study (RQ2)

To answer RQ2, we provide more detailed analysis on CrossDyG by validating the utility of its contained different components. Specifically, we compare CrossDyG with its two variants: (1) w/o Repeat, which replaces the adopted repeat sampling with the commonly used recent sampling; (2) w/o Causal, which removes the causal

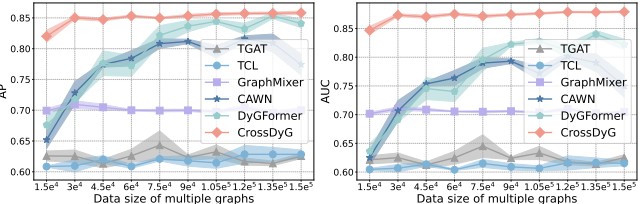

(a) Performance in terms of AP. (b) Performance in terms of AUC.
**Figure 5: Results by training on Wikipedia and Reddit of different size and evaluating on LastFM.**

mechanisms including deconfounded training to eliminate the structural bias in training and causal intervention to leverage the structural characteristics in inference. The results are shown in Table 3.

From Table 3, we can observe that CrossDyG obviously outperforms its two variants in terms of AP and AUC in different cases. This indicates the utility of the repeat sampling for removing the noisy neighbors and the causal mechanisms including deconfounded training and causal intervention for dealing with the structural characteristics during training and inference, respectively. In addition, comparing w/o Repeat and w/o Causal, we can find that the performance decreasing is generally larger when remove the repeat sampling. We analyze this is due to that the adopted repeat sampling can not only extract the most related neighbors for learning the node correlations, but also serve as a basis for deconfounded training and causal intervention as we point out in Section 3.3.4.

## 4.5 Training on Multiple Graphs (RQ3)

To answer RQ3, we adopt two small datasets Wikipedia and Reddit for training and utilize the largest dataset LastFM for evaluation. Specifically, we randomly sample the same number of dynamic links from Wikipedia and Reddit, then combine them together for model optimization. In addition, we range the sampled link number in each dataset in $\{1.5e^4, 3.0e^4, \ldots, 1.5e^5\}$ to further analyze the impact of the data size of multiple graphs on the prediction performance. The performance of CrossDyG and the baselines in terms of AP and AUC are shown in Fig. 5.

From Fig. 5, we can observe that CrossDyG outperforms the baselines when learning from different sizes of multiple graphs, indicating the robustness of CrossDyG when training on multiple graphs. Moreover, we can observe that for the baselines TGAT, TCL and GraphMixer without modeling the node correlations, the performance is stably low when the data size increases. Differently for CAWN and DyGFormer which consider the node correlations, its

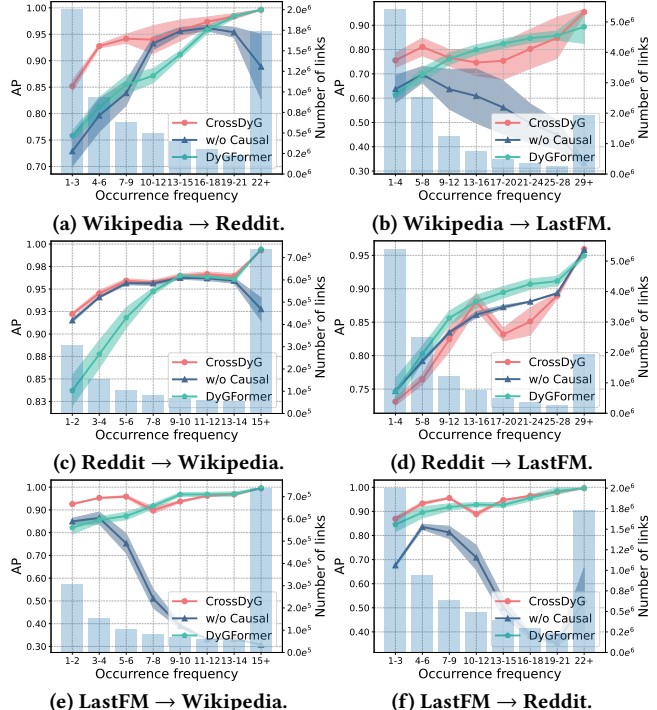

(a) Wikipedia → Reddit.

(b) Wikipedia → LastFM.

(c) Reddit → Wikipedia.

(d) Reddit → LastFM.

(e) LastFM → Wikipedia.

(f) LastFM → Reddit.

**Figure 6: Performance on different occurrence numbers.**

performance increases rapidly with the data size increasing, while the performance is especially poor with small data sizes. However, supported by the ability of learning the universal network evolution pattern, CrossDyG obtains stably outstanding performance on different data sizes and shows large performance gap between the baselines when learning from limited multiple graph data especially.

## 4.6 Effect of Structural Bias

To answer RQ4, we provide an intuitive observation of the utility of deconfounded training and causal intervention for dealing with the structural characteristics in CrossDyG, we compare the performance of CrossDyG and its variant w/o Causal as well as DyGFormer on groups containing links with various occurrence frequencies. Specifically, we divide the dynamic links in each dataset into 8 groups, where the group size is set to 2, 3 and 4 for Wikipedia, Reddit and LastFM considering their different dataset size, respectively. The results are shown in Fig. 6.

First, by comparing the variant w/o Causal with DyGFormer, we can find that on small occurrence frequencies, w/o Causal performs well and can generally outperform DyGFormer. This is due to that through the repeat neighbor sampling, w/o Causal accurately extracts the neighbors most related to the structural correlations, thus eliminating the noisy neighbors for accurate predictions. However, the performance of w/o Causal is poor on large occurrence frequencies, which is due to that w/o Causal trained on the source graph cannot well adapt to each test target graph during inference. By solving this problem using the causal mechanisms including deconfounded training and causal intervention, CrossDyG ensures outstanding performance on various occurrence frequencies, thus achieving the state-of-the-art overall performance.

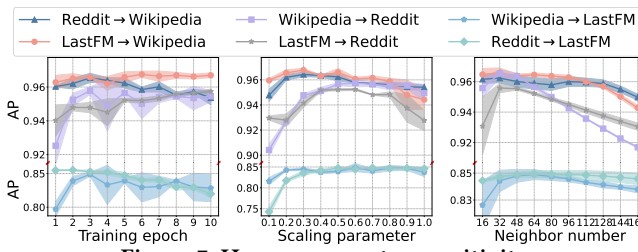

**Figure 7: Hyper-parameter sensitivity.**

## 4.7 Hyper-Parameter Sensitivity

To answer RQ5, we tune the hyper parameters of CrossDyG including the training epoch, the scaling parameter $\gamma$ and the neighbor number as stated in Section 4.2.3, to analyze the sensitivity of CrossDyG to its contained different hyper parameters. The results in terms of AP are presented in Fig. 7.

*Training epoch.* For the training epoch, we can observe that epoch = 1 can already ensures satisfactory performance in cases trained on Reddit and LastFM. Differently when trained on Wikipedia, the performance increases in the first several epochs with the pretraining epoch increasing, which may be due to its smaller dataset size than Reddit and LastFM. That is, more iterations are required for model optimization with small size of training data.

*Scaling parameter.* For the scaling parameter $\gamma$, we can see that with $\gamma$ increasing, the performance tested on Wikipedia and Reddit first increases and then obviously decreases, while the performance tested on LastFM shows a stable trend when $\gamma$ is large. This may be due to that the occurrence frequencies are generally large on the largest dataset LastFM, thus a large $\gamma$ is proper for smoothing the occurrence frequency.

*Neighbor number.* For the neighbor number, it can be observed that a small neighbor number of 32 is generally enough for achieving the optimal performance in various cases. This may be due to that large neighbor numbers may introduce noisy neighbors, which represent the specificity of each dataset, thus bringing in negative impact on transferring the universal network evolution pattern.

## 5 CONCLUSIONS AND FUTURE WORK

In this paper, we focus on improving the cross-graph transferability of dynamic link prediction in a one-many mechanism (*i.e.*, training on one single dynamic graph and testing on multiple unobserved graphs) and propose a novel framework named CrossDyG. Specifically, CrossDyG learns the universal network evolution pattern across different graphs using deconfounded training from the single source graph in training, and then adopts causal intervention to leverage the graph-specific structural characteristics of each target graph for making predictions during inference. Comprehensive experiments on three benchmark datasets of dynamic graphs validate the effectiveness of CrossDyG, with steady and significant performance improvements.

As to future work, we would like to exploit the utility of large language models (LLMs) [1, 9] for modeling the contained node/edge features in dynamic graphs. In addition, we are also interested in further improving the efficiency of cross-graph transferability using the graph condensation strategy [5, 11].

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
