# OpenReview forum: "On the Cross-Graph Transferability of Dynamic Link Prediction"
_ACM.org/TheWebConf/2025/Conference — WWW 2025 Poster_

### Official Review · Reviewer_yzf2 · 2024-11-25

**Novelty:** 4
**Technical Quality:** 4

**Review:**

The paper leverages causal deconfounding for dynamic link prediction, aiming at enhancing cross-graph transferability by mitigating structural biases.

**Pros**
- The deconfounded training method applied to the cross-graph link prediction is theoretically sound.
- The effectiveness of CrossDyG is well-supported by the experiments.

**Cons**
- Causal inference method has been applied to link prediction by several existing works for structural debiasing, e.g. [1]. Although most are targeted for in-graph tasks, applying deconfounded training to cross-graph tasks seems more of an adaptation than a fundamentally new approach.
- While the paper claims that existing works follow a many-many mechanism for cross-network prediction that is computationally expensive (in the abstract and introduction), there are no scalability experiments in this paper to support this claim.
- The evaluation is restricted to only three datasets, which seems too scarce for evaluating a one-many prediction method.

[1] Fan, S., Wang, X., Mo, Y., Shi, C., & Tang, J. (2022). Debiasing graph neural networks via learning disentangled causal substructure. Advances in Neural Information Processing Systems, 35, 24934-24946.

**Questions:**

- Although CrossDyG is applied to dynamic graphs, it seems that the methodology itself does not leverage the temporal aspect of the graph except for the fusion of temporal encoding, which can be easily excluded from the algorithm. Could you explain how CrossDyG is closely related to dynamic link prediction?
- Is it possible to include scalability results of CrossDyG, especially compared to other many-many prediction methods?
- Is it possible to include more datasets?
- (Minor Issue) In line 393, it seems that the authors are referring to the **repeated** neighbor sampling rather than the recent sampling since the follow-up sentence is referring readers to Figure 3b. I wonder if it is a typo, otherwise it is confusing.

**Reviewer Confidence:**

2: The reviewer is willing to defend the evaluation, but it is likely that the reviewer did not understand parts of the paper

**Scope:**

3: The work is somewhat relevant to the Web and to the track, and is of narrow interest to a sub-community

---

### Official Review · Reviewer_BduB · 2024-12-01

**Novelty:** 5
**Technical Quality:** 5

**Review:**

## Summary:

This paper is highly intriguing as it lays a foundational groundwork for the “one-to-many” direction in graph transferability.  This article proposes a new method called CorssDyG,  which learn the universal network evolution pattern form one single source to generalize to other unobserved graphs. and provide causal and empirical analysis on the structure bias caused by the graph-specific structural characteristics which CrossDyG eliminate.  And has achieved outstanding performance on the extensive experiments.

### Scope: 4

This paper focuses on dynamic link prediction of graphs, which is highly relevant to the Web domain.



### Novelty：5

This paper addresses the one-to-many transferability problem, a direction that has not been explored previously, and introduces the innovative use of **do-calculus**, demonstrating significant originality.



### Technical Quality：5

This paper utilizes three datasets and compares its approach against multiple baseline methods. The experimental results demonstrate the effectiveness of the proposed method, and ablation studies further validate its contributions. However, including a comparison with the DyExpert method would make the findings more compelling.

**Questions:**

- Why is Neighbor Sequence Sampling utilized in Section 3.2.1, and what would be the potential impact of using the entire original graph instead?
- In Section 3.3, Analysis of Structural Bias, where is your citation? How does your reasoning align with the representation shown in Figure 2?
- In Section 4.2.2, Baselines for Compariso,  why is DyExpert not included as a baseline? Can the many-to-many mechanism be transformed into a one-to-many mechanism?

**Reviewer Confidence:**

3: The reviewer is confident but not certain that the evaluation is correct

**Scope:**

4: The work is relevant to the Web and to the track, and is of broad interest to the community

---

### Official Review · Reviewer_PLVY · 2024-12-01

**Novelty:** 5
**Technical Quality:** 4

**Review:**

Pros:

1. The paper proposes an effective model for one-to-many cross-graph link prediction, grounded in the general insight that link prediction is a generalized task across different domains.
2. The authors conduct extensive evaluations and provide the source code.
----

Cons:

1. The motivation for the proposed approach is not sufficiently evaluated or validated. Intuitively, while DyGFormer and GraphMixer were originally designed for in-graph link prediction, they still achieve competitive performance in one-to-many settings. From this perspective, it seems that one-to-many cross-graph LP is not difficult. A more thorough discussion and analysis of this aspect would strengthen the paper.

2. The experimental results are not fully explained and some phenomena are a bit confusing. For example, (1) In Table 3 of the ablation study, both variants are significantly worse than DyGFormer and GraphMixer, which is just designed for base Link Prediction. (2) The results of the ablation experiment have huge variance on different data sets. For *Reddit-->Wiki, LastFM*, the casual module is somewhat ineffective. (3) The detailed settings of some experiments are unclear (e.g., Section 4.5). A more in-depth and careful analysis would be beneficial.

3. The paper does not include an efficiency analysis. Particularly, the primary advantages of the one-to-many setting is the potential reduction in computational costs. It would be beneficial to provide a detailed performance-to-efficiency ratio (Performance/Efficiency) to highlight this advantage.

4. The paper does not consider the cross-domain features in one-to-many settings. For example, LastFM dataset lacks node or edge features, and the authors should explain the feature alignment for processing such cases. This is especially critical given the inherent challenges posed by attribute heterogeneity across domains.

5. The experimental evaluation does not include some widely-used datasets, such as the MOOC dataset. Given the performance variance observed in Table 1, including additional datasets would provide more comprehensive validation. At a minimum, benchmark datasets like MOOC and UN-Vote should be considered.

6. The contributions of this paper to the Social Media is not clear. The work appears to focus on graph learning, making it potentially more suitable for the Graph Track.

**Questions:**

Q1: Could the one-to-many setting be considered a special case of many-to-many? While one-to-many appears promising, it seems less feasible in scenarios of extreme heterogeneity in attributes and structure. Many-to-many settings, which aim to learn commonalities from a larger data pool, appear more robust and generalized. Does one-to-many achieve similar effects, and if so, could the authors please introduce how?

Q2: Apart to reducing the cost of training multiple graphs, what specific advantages does the one-to-many setting offer? In Fig.5, the performance of the many-to-many setting is *significantly worse* than the one-to-many setting, which is very unexpected. Could the authors clarify why the one-to-many approach, which is theoretically more challenging, outperforms the many-to-many setting significantly? Could the authors introduce the experimental setup for training on multiple graphs?

Q3: Does using repeat sampling instead of recent sampling result in a degradation of dynamic modeling capability? The modeling of dynamics currently relies on temporal encoding, but this raises concerns on dynamics. By sacrificing some temporal information to align data distributions across datasets, how can we ensure that valuable information is not lost? Are there theoretical or empirical justifications for this trade-off?

**Reviewer Confidence:**

4: The reviewer is certain that the evaluation is correct and very familiar with the relevant literature

**Scope:**

2: The connection to the Web is incidental, e.g., use of Web data or API

---

### Official Review · Reviewer_QPeF · 2024-12-02

**Novelty:** 4
**Technical Quality:** 3

**Review:**

The paper proposes CrossDyG, a new framework for cross-graph dynamic link prediction, aiming to transfer knowledge from one source graph to multiple target graphs. It introduces deconfounded training to eliminate structural bias and uses causal intervention to adapt predictions to graph-specific characteristics. Their experiments demonstrate that CrossDyG outperforms state-of-the-art baselines in terms of both AP and AUC, especially when trained on small graphs.

**Strength**

- The paper proposes a one-to-many mechanism for cross-graph dynamic link prediction, addressing a gap in existing methods.
- The results show clear performance improvements, with up to 17% gains in AUC over baselines.

**weaknesses**

- The paper mentions computational efficiency but lacks a detailed analysis of runtime or memory usage compared to other methods.
- While Table 1 presents performance metrics for different source-target graph pairs, the underlying reasons for the varying results remain unclear. For example, why does Wikipedia → Reddit outperform Wikipedia → LastFM, and vice versa? A dedicated discussion section exploring the characteristics of source and target graphs (e.g., density, temporal dynamics, or structural patterns) that contribute to better transferability would significantly enhance the paper.
- The current experiments focus on moderately sized graphs, but it remains unclear how CrossDyG performs on extreme cases, such as very sparse or very dense graphs. Evaluating CrossDyG on such graphs would be useful. A potential dataset for sparse graph evaluation could be Social Evolution, as provided in [35].

**Typos**

-In section 3.3.4, line 382, "trasferability"-> "transferability."

**Questions:**

Please see weaknesses

**Reviewer Confidence:**

2: The reviewer is willing to defend the evaluation, but it is likely that the reviewer did not understand parts of the paper

**Scope:**

3: The work is somewhat relevant to the Web and to the track, and is of narrow interest to a sub-community

---

### Official Review · Reviewer_KZ6x · 2024-12-02

**Novelty:** 5
**Technical Quality:** 5

**Review:**

Quality:
The paper introduces CrossDyG, a novel approach addressing cross-graph dynamic link prediction. The one-many mechanism, where a model is trained on a single source graph and tested on multiple target graphs, is innovative and relevant for real-world applications. The methodology, which incorporates causal analysis and deconfounded training, is rigorous and supported by empirical experiments on benchmark datasets. The paper answers the following five research questions:
    - RQ1: Can CrossDyG improve cross-graph transferability across different graphs in a one-many mechanism by learning from one single source dynamic graph?
    - RQ2: What is the contribution of the different components of CrossDyG to prediction performance?
    - RQ3: How does CrossDyG compare with baselines when training on multiple source dynamic graphs?
    - RQ4: Can CrossDyG mitigate the bias of structural characteristics during training and leverage them in inference?
    - RQ5: How sensitive is CrossDyG’s performance to its hyperparameters?
The well-defined research questions make the paper consistent and focused.


Clarity:
The paper is well-organized, easy to follow, and effectively communicates its contributions. The methodology and results are clearly explained and comprehensible.


Originality:
The concept of achieving cross-graph transferability using a one-many mechanism, coupled with the integration of causal analysis to address structural bias, is a significant contribution that distinguishes this work from existing methods.


Significance:
This work addresses a critical limitation in current dynamic link prediction methods, namely their inability to generalize to unobserved graphs. By achieving significant improvements in AP and AUC, especially on small source graphs, CrossDyG demonstrates strong potential for applications in recommender systems, social networks, and other domains involving dynamic graphs, particularly web graphs.

Pros:
    1. Innovative Approach: The one-many mechanism and causal-based deconfounded training are significant contributions.
    2. Strong Empirical Results: Demonstrates substantial improvements over state-of-the-art baselines, with gains of up to 17.02% in AUC.
    3. Sound Theoretical Foundations: The causal analysis provides a solid theoretical basis for the methodology.
    4. Practical Applicability: The approach is computationally efficient and applicable to various real-world scenarios involving dynamic graphs.
Cons:
    1. Scalability Discussion: There is no discussion on how CrossDyG scales with increasing graph size and complexity. This is particularly important as one of the paper’s main motivations was to address the high computational cost of existing dynamic graph foundation models (e.g., DyExpert). A deeper exploration of the scalability of CrossDyG compared to baseline models would strengthen the work.

    2. Limited Experimentation: The paper primarily relies on real-world datasets for experimentation. However, to validate the results and ensure their generality beyond the three datasets used, the authors could consider testing on artificial graphs. For example, generating dynamic graphs using the dynamic LFR model (  Greene, D., Doyle, D., & Cunningham, P. (2010, August). Tracking the evolution of communities in dynamic social networks. In 2010 international conference on advances in social networks analysis and mining (pp. 176-183). IEEE) could provide valuable insights. The authors could:
        - Analyze the impact of communities: Generate a dynamic graph with a small mixing parameter (μ=0.1) and use it as the source graph for training. Then, generate two additional graphs with larger mixing parameters (μ=0.5 and μ=0.9) for testing. This setup would illustrate the one-many mechanism and clarify how the presence of strong communities affects performance.
        - Explore the effect of event types: Generate graphs with fixed dynamic events (e.g., intermittence, merging, splitting) for training, and use graphs with different events for testing. This would help evaluate the robustness of CrossDyG across various dynamic scenarios.
        - Scalability Testing: Use the same dynamic graph model and analyze the impact of graph size by varying its parameters, such as the number of nodes and edges.

**Questions:**

1. Scalability: What is the computational complexity of the proposed approach, and how does it scale with large graphs and datasets?
    2. Experimentation Could you provide additional experiments with artificial graphs to better demonstrate the robustness of the proposed approach?

**Reviewer Confidence:**

3: The reviewer is confident but not certain that the evaluation is correct

**Scope:**

4: The work is relevant to the Web and to the track, and is of broad interest to the community